

# Repeatedly adopting power postures does not affect hormonal correlates of dominance and affiliative behavior

Hannah Metzler[1,2] and Julie Grèzes[1]

[1] Laboratoire de neurosciences cognitives et computationnelles, INSERM U960, Département d'études cognitives, Ecole Normale Supérieure de Paris, PSL University, Paris, France
[2] Sorbonne Universités, Université Pierre et Marie Curie (Paris VI), Paris, France

## ABSTRACT

**Background:** Adopting expansive vs. constrictive postures related to high vs. low levels of social power has been suggested to induce changes in testosterone and cortisol levels, and thereby to mimic hormonal correlates of dominance behavior. However, these findings have been challenged by several non-replications recently. Despite this growing body of evidence that does not support posture effects on hormone levels, the question remains as to whether repeatedly holding postures over time and/or assessing hormonal responses at different time points would yield different outcomes. The current study assesses these methodological characteristics as possible reasons for previous null-findings. Additionally, it investigates for the first time whether expansive and constrictive postures impact progesterone levels, a suggested correlate of affiliative motives and behavior. By testing the effects of repeated but short posture manipulations in between the blocks of a social task while using a cover story, it further fulfills the conditions previously raised as potentially necessary for the effects to occur.
**Methods:** A total of 82 male participants repeatedly adopted an expansive or constrictive posture for 2 min in between blocks of a task that consisted in categorizing faces based on first impressions. Saliva samples were taken at two different time points in a time window in which hormonal responses to stress, competition and other manipulations are known to be strongest.
**Results:** Neither testosterone and cortisol levels linked to dominance behaviors, nor progesterone levels related to affiliative tendencies, responded differently to adopting expansive as opposed to constrictive postures. The present results suggest that even repeated power posing in a context where social stimuli are task-relevant does not elicit changes in hormone levels.

Corresponding authors
Hannah Metzler,
hannahmetzler1@gmail.com
Julie Grèzes, julie.grezes@ens.fr

## INTRODUCTION

Individuals' position in a social hierarchy greatly determines their response to stressful situations as well as their opportunities for social contact and relationships (*De Waal, 1986*; *Sapolsky, 2005*). Because individuals' social power changes over time and across different contexts, the physiological mechanisms underlying power-related behavior need to

allow flexible adaptation to new situations. Steroid hormone levels, including cortisol, testosterone and progesterone are key players in the implementation of this behavioral flexibility: not only do their baseline levels influence individuals' tendencies for certain behaviors, but their levels also change in situations that involve stress, opportunities for gaining social status or affiliating with others, or threats to social status and affiliative needs (*Mehta & Josephs, 2011*; *Schultheiss, 2013*). Although there are complex interactions between these three steroid hormones and the behaviors they modulate (*Mehta & Josephs, 2011*), cortisol is predominantly involved in the regulation of stress responses (*Sapolsky, 1990*), testosterone seems to mediate behaviors that serve to achieve or maintain social status (*Archer, 2006*; *Mehta & Josephs, 2011*; *Eisenegger, Haushofer & Fehr, 2011*; *Nave et al., 2018*), and progesterone has been suggested to contribute to the regulation of affiliative behavior (*Schultheiss, Wirth & Stanton, 2004*; *Wirth, 2011*).

Positions of high and low power are associated with distinct endocrine profiles: whereas high-ranking individuals have higher baseline testosterone levels and lower cortisol levels, the reverse is observed in low-ranking individuals (*Sapolsky, 1990*; *Virgin & Sapolsky, 1997*; *Mehta & Josephs, 2010*). Building on theories of embodiment, which postulate that many aspects of cognition are shaped by representations of body actions, *Carney, Cuddy & Yap (2010)* assessed whether exhibiting non-verbal dominant or submissive behavior, namely expanding or constricting one's body, would induce corresponding changes in testosterone and cortisol levels. They did indeed observe an increase of testosterone and a decrease of cortisol in individuals who had adopted an expansive posture and the reverse changes in individuals who had adopted a constrictive posture. Although these findings seemed consistent with the hormonal correlates of status and power, four subsequent studies could not replicate them despite large sample sizes that ensured high statistical power in three of the replications (*Ranehill et al., 2015*; *Ronay et al., 2017*; *Smith & Apicella, 2017*; *Davis et al., 2017*). The first replication (*Ranehill et al., 2015*) exhibited small but potentially crucial methodological differences with the original study (see *Carney, Cuddy & Yap, 2015*). Two others attempted to improve the original study's setting by testing effects in more ecologically valid social contexts that bear implications for power, status and dominance, such as competition or public speaking (*Smith & Apicella, 2017*; *Davis et al., 2017*). Only *Ronay et al.'s (2017)* study was a direct replication of the original study by *Carney, Cuddy & Yap (2010)*, but also observed no significant effects.

All these studies focused only on power-related behavior and hormones. Yet, there is considerable evidence that power also impacts individuals' affiliative tendencies (*Magee & Smith, 2013*; *Guinote, 2017*). For instance, lack of power enhances motivation to connect with others (*Lammers et al., 2012*; *Case, Conlon & Maner, 2015*) and cues of low social status have positive effects on pro-social behavior (*Guinote et al., 2015*). Moreover, facing threats and stressful situations can enhance affiliative motivation and behavior (*Schachter, 1959*; *Gump & Kulik, 1997*; *Dezecache, Grèzes & Dahl, 2017*), as bonding with others represents an efficient coping strategy (*Taylor, 2006*; *Dezecache, 2015*). The display of constrictive and submissive postures generally occurs in threatening situations and serves to appease aggressive conspecifics by signaling friendly intentions (*Schenkel, 1967*; *De Waal, 1986*). Adopting constrictive postures may thus be linked with affiliative

tendencies. Progesterone is known to be released together with cortisol in response to stress in general, but particularly social stress (*Wirth, 2011*). It correlates with both naturally fluctuating (*Schultheiss, Dargel & Rohde, 2003*) and experimentally induced affiliative motivation (*Schultheiss, Wirth & Stanton, 2004*; *Wirth & Schultheiss, 2006*) and may promote social bonding as a coping behavior in response to stress (*Wirth, 2011*). The question remains as to whether adopting constrictive postures would lead to a change in salivary progesterone, thereby indicating an increase of affiliation motivation.

Although expansive and constrictive postures' effects on cortisol and testosterone were not replicated, their effect on feelings of power and control, also originally reported by *Carney, Cuddy & Yap (2010)*, has been confirmed by a meta-analysis (*Gronau et al., 2017*) of six pre-registered and highly powered studies (*Ronay et al., 2017*; *Bombari, Mast & Pulfrey, 2017*; *Jackson et al., 2017*; *Keller, Johnson & Harder, 2017*; *Klaschinski, Schnabel & Schröder-Abé, 2017*; *Bailey, LaFrance & Dovidio, 2017*). On the basis of a *p*-curve analysis, *Cuddy, Schultz & Fosse (2018)* further suggested that the literature existing prior to these pre-registered studies also contains evidence for an effect on feelings of power. It should be noted that this analysis confounded significant posture effects in opposite directions (*Credé, 2018*), and further included several studies that did not satisfy the criteria for a robust and reproducible *p*-curve result (*Simonsohn, Nelson & Simmons, 2017*). Nevertheless, and despite previous null-findings for hormones, the evidence for posture effects on feelings of power and other emotional and affective self-report measures in their *p*-curve analysis have led *Cuddy, Schultz & Fosse (2018)* to call for more studies on psychophysiological outcomes. Specifically, they suggest that future experiments should apply more precise hormone-measurement methods or assess the incremental effects of adopting a posture several times. *Davis et al. (2017)* have also raised the question of whether null-effects for hormones could be related to the timing and dose of the posture manipulation. They speculated that larger doses of posture or collection of samples at different time points after the posture could yield different outcomes. Indeed, adopting expansive postures for about 5 min throughout a stressful experience boosted the cortisol response to stress (*Turan, 2015*). This suggests that expansive postures are maladaptive in certain contexts, but also illustrates that adopting postures for longer durations may induce hormonal changes.

Altogether, it appears that additional empirical evidence is necessary to reach final conclusions about whether expansive and constrictive postures induce changes in testosterone or cortisol levels at different time points than assessed previously or when adopted for longer durations. In addition, it remains unexplored whether these postures affect hormonal correlates of affiliative behavior, such as progesterone. In 2015, before the publication of the first non-replication of the power posing effect on hormone levels (*Ranehill et al., 2015*), we conducted a study which we believe can contribute to the ongoing discussion about whether expansive and constrictive postures induce changes in testosterone or cortisol levels, and that additionally assessed progesterone levels for the first time. It provides an answer to questions regarding the timing of hormone measurements and the "dose" of posture recently raised by *Cuddy, Schultz & Fosse (2018)* and *Davis et al. (2017)*. First, hormone levels were measured at longer time intervals after

the start of the posture manipulation than in previous studies. Second, incremental posture effects were examined by having participants repeatedly adopt an expansive or constrictive standing posture in between the blocks of a face categorization task. Additionally, participants were encouraged to adopt a freely adaptable open or closed sitting posture throughout the task.

The study's design met the criteria which *Carney, Cuddy & Yap (2015)* pointed out as potentially necessary conditions for posture effects in their response to the first non-replication. Specifically, the study's procedure included (1) a credible cover story, (2) instructions delivered by an experimenter instead of a computer, (3) short time windows for adopting the postures in order to avoid discomfort and (4) a face categorization task resembling the social filler task in the original study (*Carney, Cuddy & Yap, 2010*). The original social filler task consisted of forming impressions of faces without providing a response, and is commonly interpreted as a "social context" in the posture literature (*Cesario & McDonald, 2013*; *Carney, Cuddy & Yap, 2015*). Aiming for a stronger emphasis on social aspects, we made faces task-relevant, having participants categorize them according to their implicitly assigned minimal group membership. Knowing that affective state and dominance-related personality traits can influence individuals' body posture (*Weisfeld & Beresford, 1982*; *Canales et al., 2017*; *Aviezer, Trope & Todorov, 2012*), and may therefore modulate hormonal responses to transiently adopted postures, we collected several state and trait self-report measures to check for any potentially relevant differences between the two posture groups, and control for these if necessary. In summary, the current study investigated changes in salivary testosterone, cortisol and progesterone levels in response to a repeated expansive or constrictive posture manipulation in the context of a face categorization task.

## MATERIALS AND METHODS

### Participants

*Carney, Cuddy & Yap (2010)* reported effect-sizes of $r = 0.34$ for testosterone and $r = 0.43$ cortisol. We performed a power-analysis in G-Power (*Faul et al., 2007*) based on the smaller one of these two effect-sizes, that is, $r = 0.34$. This yielded a minimal necessary sample of $n = 63$ to achieve 80% power to detect effects as large as those of *Carney, Cuddy & Yap (2010)*. These were the only available effect sizes for posture effects on hormone levels when we conducted our study. Given inherent biological differences in testosterone and progesterone production between men and women, analyses of these hormones need to be done separately for each sex (*Stanton, 2011*). Therefore, we included only male participants to achieve sufficient power with the maximum sample size possible under our feasibility constraints.

We recruited a total of 82 male participants via a participant pool mailing list and student job advertisement websites. Participants were between 17 and 32 years old, reported not to be regular smokers or under medical treatment, and to not have a history of endocrine illness, neurological and psychiatric disorders, or dependency to alcohol or other drugs. All participants provided written informed consent and were paid for their participation. The experimental protocol was approved by INSERM and licensed

by the local research ethics committee (Comité de protection des personnes Ile de France III—Project CO7-28, N° Eudract: 207-A01125-48) and carried out in accordance with the Declaration of Helsinki.

It should be noted that our sample size was estimated with a power-analysis based on an effect size ($r = 0.34$), which is unrealistically large given the evidence acquired since 2010. Repeating the same power analysis based on a more realistic estimate of $r = 0.10$ (in line with the effect on feelings of power, see *Gronau et al., 2017*) yields a much larger required sample size of 779. Given that our sample size was much smaller, we conducted a sensitivity analysis to estimate the smallest effect size we could detect with 80% power. This further allowed considering specificities of our design, such as the three time points of saliva collection. The sensitivity analysis in G*Power for a between (two groups) by within (three time points) interaction with 82 participants suggested that we had 80% power to detect effect sizes larger than partial eta-squared = 0.06.

### Measures

#### Questionnaires

For assessing potential differences between the posture groups, we administered a collection of self-report questionnaires. Because power, as opposed to lack of power, decreases anxiety and inhibition, increases confidence, self-esteem and approach motivation in general (*Keltner, Gruenfeld & Anderson, 2003*; *Guinote, 2017*), we included the French version of the State-Trait Anxiety Inventory (STAI, *Gauthier & Bouchard, 1993*), the Rosenberg Self-Esteem Scale (*Vallières et Vallerand, 1990*), and the BIS/BAS scales which assess the behavioral activation and inhibition systems (*Caci, Deschaux & Baylé, 2007*). Participants completed the trait measures prior to the testing session in the laboratory but filled out the state version of the STAI after arrival at the laboratory. In addition, questions regarding compliance with behavioral restrictions before saliva collection and the dominance scale from the International Personality Item Pool (*Goldberg et al., 2006*, scale representing the California Psychological Inventory: https://ipip.ori.org/newCPIKey. htm#dominance) were administered at the end of the experiment to avoid raising suspicion about the real purpose of the posture manipulation.

#### Saliva collection

We collected three saliva samples (one ml each) per participant using small tubes and stored them below −20 °C immediately after collection. After completion of the study (duration: 51 days), they were packed in dry ice and shipped to the laboratory of Clemens Kirschbaum in Dresden, where they were analyzed with commercially available chemiluminescence immunoassays with high sensitivity (IBL International, Hamburg, Germany). For a more detailed description of the assay methods used by this laboratory, see for example *Ronay et al. (2017)*. To exclude the possible influence of external factors on hormone levels, participants were requested to refrain from drinking alcohol and exercising intensively within 24 h before the session, from smoking or taking medical drugs on the testing day, and from eating, drinking anything except water, and tooth brushing 1.5 h before the session. The debriefing questionnaire after the experiment

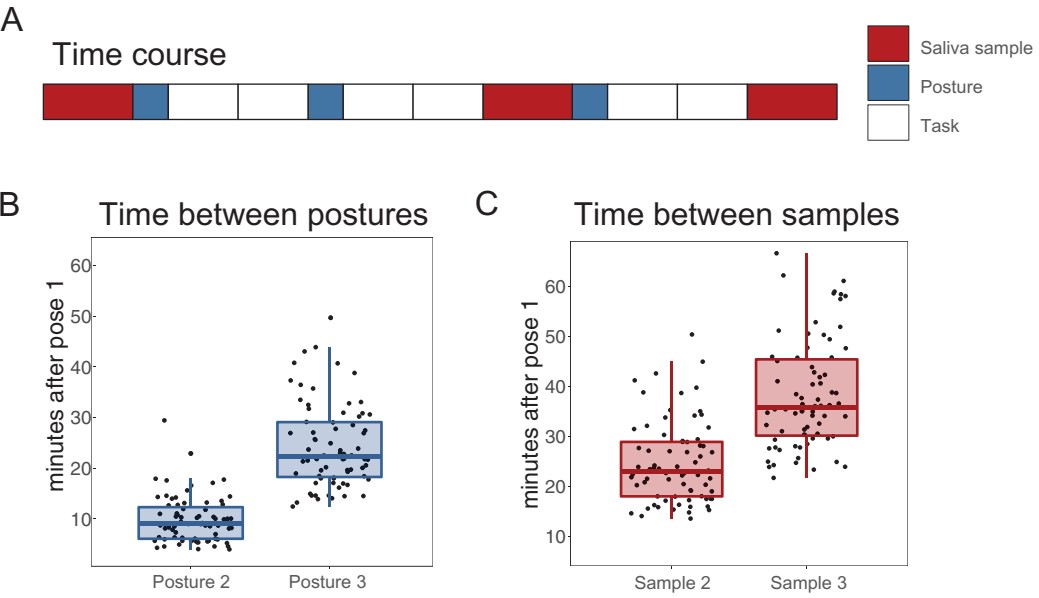

**Figure 1 Time course of the experiment and adopted body postures.** (A) Time course of postures, saliva sample and task blocks. Participants adopted the same posture three times, before every other block of a face categorization task. Before the first posture, and after block 4 and 6, they provided a saliva sample. (B) Time from the beginning of posture 1 to posture 2 and 3. (C) Time from the beginning of posture 1 to saliva sample 2 and 3.

showed that they largely complied with these instructions (five exceptions for alcohol, two for smoking).

## Procedure

All testing sessions took place between 13 and 19 h to attenuate effects of diurnal variation of hormone levels. Saliva samples were collected as part of another study (*Metzler, 2018*, see Study 1 in Chapter 6) during which participants had to categorize faces into in- and outgroup members while repeatedly adopting postures between task blocks. Figures 1A and 1B depict the time course of postures, saliva samples and interleaved blocks of the face categorization task.

Upon arrival, participants signed consent forms and completed the STAI state questionnaire. Following the example of a previous study (*Ratner et al., 2014*), we implicitly assigned participants to one of two arbitrary groups using a traditional minimal group procedure (*Tajfel, 1970*). Participants' task was to choose either their in- or outgroup members from pairs of faces based on their first impression. This setting provided a credible cover story, namely that the saliva samples were collected to assess associations between face categorization and physiological indices. The cover story for the postures was that a second, unrelated project on the impact of body posture on heart rate was conducted simultaneously. Participants would adopt the posture three times for 2 min each time in between the blocks of the face categorization task. This supposedly served to acquire heart-rate data for a total of 6 min while avoiding discomfort from holding the same posture for too long, and offered breaks during the visually demanding task. At this

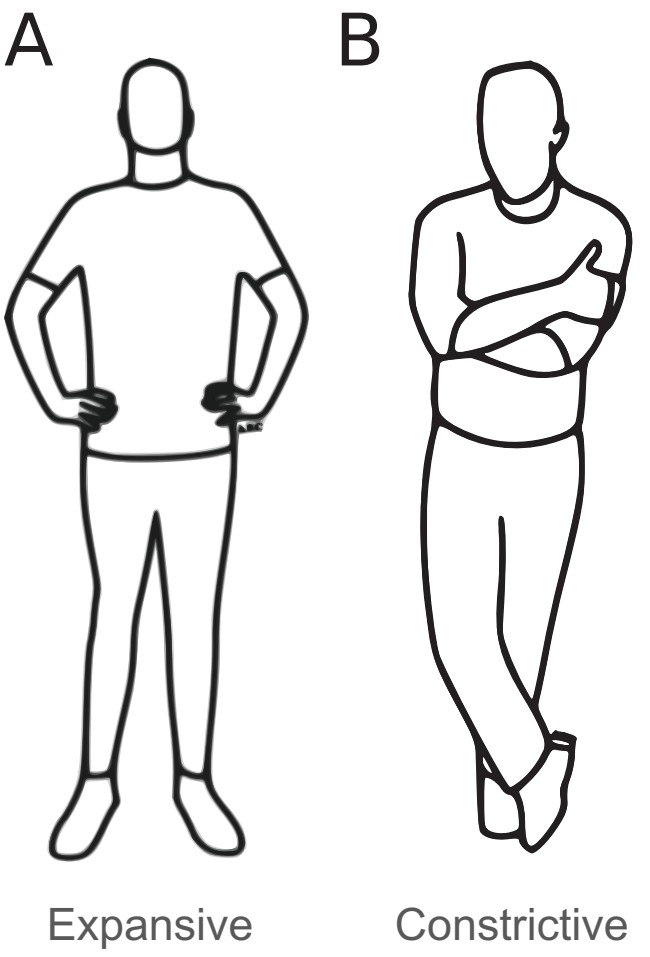

A  B

Expansive  Constrictive

**Figure 2  Adopted postures.** Postures adopted by the two experimental groups. (A) Expansive posture. (B) Constrictive posture. Images created by Antoine Balouka-Chadwick.

point of the instructions, approximately 15 min after arrival, participants provided a first saliva sample.

Thereafter, the female experimenter determined the posture condition using a randomizing function and provided corresponding instructions for either the expansive ($n = 42$) or the constrictive ($n = 40$) posture. Up to the point where posture instructions were provided, that is, at the end of all other procedures, the experimenter was blind to participant's posture in order to minimize possible experimenter biases. She first placed electrodes on participant's wrists and hooked them up to the acquisition system, and demonstratively turned it on. Next, she verbally provided instructions on how to place each body part without demonstrating the posture herself. The expansive and constrictive posture involved open or closed limbs, erect or slumped upper body and straight or downward head tilt, respectively (see Fig. 2). The experimenter informed participants that she would check whether they correctly adopted this standing posture each time via a camera. Depending on the participant's posture condition, she finally instructed participants to (1) sit upright with feet apart or (2) keep back and shoulders slumped

and legs parallel or crossed during the task as far as comfortable for them, which supposedly served to "stabilize" the effect of postures on heart rate. This short instruction for the sitting posture was repeated on screen at the beginning of each task block. Although allowing participants to freely adjust their posture for their own comfort during the task constitutes a less controlled posture manipulation, it ensures higher ecological validity, as it corresponds to what we typically do in everyday life. Together, the repeated 2 min periods in which participants adopted one of two standing postures, together with the encouragement of a similar, but freely adaptable sitting position during the face categorization task, added up to a "larger dose" of posture while avoiding discomfort. Participants were alone while they adopted the postures and performed the task. The experimenter only briefly re-entered the room for the collection of two more saliva samples.

In total, participants thus adopted the standing posture three times, that is, before task block 1, 3 and 5. Saliva samples were collected before the first posture and block and after block 4 and 6. Participants had thus adopted the posture twice before sample 2, and three times before sample 3. Median block duration was 4.58 min (interquartile range (3.46–6.25)) depending on participants' speed in the face categorization task. This resulted in collection of saliva samples 2 and 3 approximately 23 and 36 min after the first posture, respectively, although the exact timing varied between participants (min. 14 min, max. 50). This corresponds to the collection of samples 2 and 3 approximately 11 and 24 min after the second posture, respectively, and collection of sample 3 approximately 10 min after the third posture.

At the end of the experiment, participants were carefully debriefed regarding suspicions about the postures. None of them had suspected a link between the posture manipulation and the saliva samples and only one participant raised doubts about our interest in a posture effect on heart-rate. Excluding him from analyses did not affect the results.

## Data analysis

Outliers were determined per time point and hormone using a conservative threshold of three times the absolute deviation from the median (*Leys et al., 2013*), given that mean ± SD rules are problematic for endocrine data which are rarely normally distributed (*Pollet & Van der Meij, 2017*).

First, we excluded one participant from all time points and hormones due to extreme progesterone values (around 1,500 pg/ml, outside of normal range even for women, see *Liening et al., 2010*), clearly indicating a problem with his salivary samples. Within the remaining sample of 81 participants (age 21.36 ± 2.78, expansive *n* = 41, constrictive *n* = 40), there were six outliers above the median plus three absolute deviations for cortisol, seven for testosterone and nine for progesterone. Results calculated without outliers did not differ from results with the full sample (see Table S1), that is, the same effects yielded significant or non-significant *p*-values with and without outlier exclusion.

All hormone levels were log-transformed to correct for right-skewed distributions and subjected to a mixed-effects ANOVA with posture (expansive, constrictive) as a between-subject and time (T1, T2, T3) as a within-subject factor. Even when the

**Table 1 Descriptive statistics for raw values of cortisol, testosterone and progesterone in samples without outliers.**

| | Posture | n | Sample | Mean | Median | SD | 95% CI |
|---|---|---|---|---|---|---|---|
| Cortisol pg/ml | Expansive | 37 | 1 | 2,755.88 | 2,599.13 | 1,416.89 | [2,299.33–3,212.44] |
| | | | 2 | 1,788.89 | 1,555.13 | 930.94 | [1,488.92–2,088.86] |
| | | | 3 | 1,421.20 | 1,268.75 | 665.40 | [1,206.79–1,635.60] |
| | Constrictive | 39 | 1 | 2,518.72 | 2,562.88 | 1,453.86 | [2,062.43–2,975.02] |
| | | | 2 | 1,647.33 | 1,638.50 | 775.28 | [1,404.01–1,890.65] |
| | | | 3 | 1,428.06 | 1,439.13 | 608.28 | [1,237.16–1,618.97] |
| Testosterone pg/ml | Expansive | 38 | 1 | 73.52 | 69.05 | 31.96 | [63.35–83.68] |
| | | | 2 | 68.37 | 60.95 | 25.90 | [60.13–76.60] |
| | | | 3 | 63.23 | 59.60 | 23.32 | [55.81–70.64] |
| | Constrictive | 37 | 1 | 78.51 | 79.30 | 30.14 | [68.79–88.22] |
| | | | 2 | 69.64 | 67.90 | 27.59 | [60.74–78.53] |
| | | | 3 | 64.92 | 62.30 | 24.97 | [56.87–72.96] |
| Progesterone pg/ml | Expansive | 38 | 1 | 55.93 | 51.20 | 29.89 | [46.42–65.43] |
| | | | 2 | 43.43 | 43.35 | 17.76 | [37.79–49.08] |
| | | | 3 | 36.69 | 38.95 | 12.59 | [32.69–40.70] |
| | Constrictive | 35 | 1 | 47.70 | 45.30 | 27.87 | [38.47–56.94] |
| | | | 2 | 39.97 | 34.30 | 22.63 | [32.47–47.46] |
| | | | 3 | 34.49 | 31.20 | 20.05 | [27.84–41.13] |

Note:
Confidence intervals are between-subject to allow for between-posture comparisons.

time * posture interaction was not significant, we performed between-posture $t$-tests on the change T1-T2 or T2-T3, to allow for comparison with previous studies with only two time points. As partial eta-squared differs between within- and between-subject designs, we additionally report generalized eta-squared as an effect-size. The latter allows for comparison between different designs, as it includes within-subject variance, and excludes variance due to other factors in the design (*Lakens, 2013*). All analysis were done in *R Core Team (2018)* using the packages ez, psych, latticeExtra, ggplot2 and dplyr (*Wickham, 2009*; *Lawrence, 2016*; *Sarkar & Andrews, 2016*; *Revelle, 2017*; *Wickham et al., 2017*). Data and analysis scripts are available at https://osf.io/3nrsy/.

## RESULTS

All hormone levels are reported in pg/ml. Descriptive statistics for raw levels of cortisol, testosterone and progesterone separated per posture and time point, including confidence intervals (CI), are presented in Table 1. The results are depicted in Fig. 3. Confidence intervals reported with $t$-tests, which were all done on log-transformed data, are log-transformed values.

### Cortisol

Cortisol levels similarly decreased over time ($F(2, 148) = 79.40$, $p < 0.001$, $\eta_{p}^2 = 0.51$, $\eta_{G}^2 = 0.16$) in both posture groups (time * posture: $F(2, 148) = 1.17$, $p = 0.313$, $\eta_{p}^2 = 0.02$,

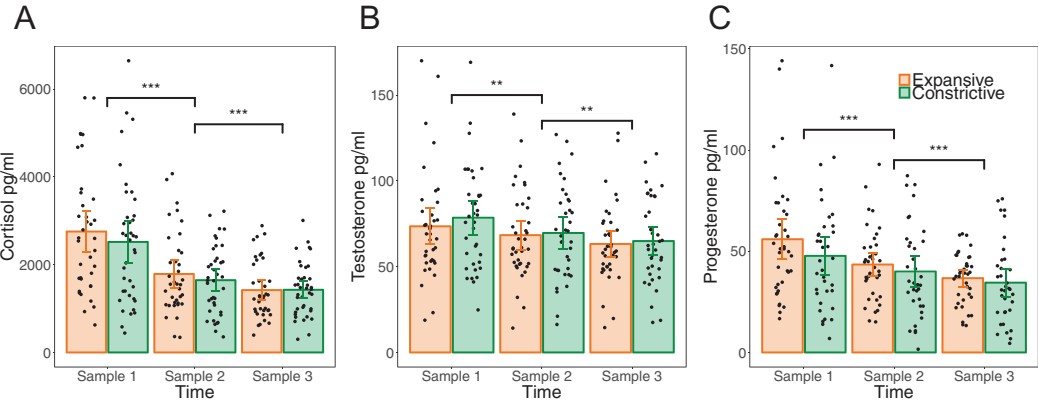

**Figure 3  Changes in hormone levels from before to after the posture manipulation.** Means, between-subject confidence intervals and individual data points for raw values of all hormone samples in pg/ml. Sample 1 was collected before the first posture. Sample 2 and 3 reflect the effect of adopting the same posture two and three times, respectively. Asterisks indicate significance in $t$-tests between time points at ***$p < 0.001$ and **$p < 0.01$. (A) Cortisol. (B) Testosterone. (C) Progesterone.

**Table 2  $T$-tests between posture groups and confidence intervals for hormone differences between time points in samples without outliers.**

|  | $t$ | d$f$ | $p$ | $d$ | 95% CI Expansive | 95% CI Constrictive |
|---|---|---|---|---|---|---|
| Cortisol T2-T1 | −0.73 | 74 | 0.470 | −0.17 | [−0.54 to −0.33] | [−0.48 to −0.27] |
| Cortisol T3-T2 | −1.02 | 74 | 0.310 | −0.23 | [−0.32 to −0.07] | [−0.22 to −0.01] |
| Testosterone T2-T1 | 1.32 | 73 | 0.190 | 0.30 | [−0.14–0.02] | [−0.21 to −0.06] |
| Testosterone T3-T2 | −0.21 | 73 | 0.840 | −0.05 | [−0.13 to −0.02] | [−0.13–0.00] |
| Progesterone T2-T1 | 0.10 | 71 | 0.920 | 0.02 | [−0.32 to −0.10] | [−0.37 to −0.07] |
| Progesterone T3-T2 | −0.32 | 71 | 0.750 | −0.07 | [−0.24 to −0.06] | [−0.24 to −0.02] |

Note:
   $T$-tests were performed on log-transformed values. Confidence intervals are between-subject to allow for between-posture comparisons.

$\eta_G^2 = 0.00$), in the absence of any overall difference between the groups ($F(1, 74) = 0.32$, $p = 0.576$, $\eta_p^2 = 0.00$, $\eta_G^2 = 0.00$). Both the decrease from T1 to T2, that is, from before the first posture to after adopting the posture twice, and the decrease from T2 to T3, that is, from after the first two postures to after the third posture, were significant (T1-T2: $t(75) = −10.67$, $p < 0.001$, $d_z = −1.22$, 95% CI [−0.48 to −0.33]), T2-T3: $t(75) = −3.78$, $p < 0.001$, $d_z = −0.43$, 95% CI [−0.24 to −0.07]). The changes T1-T2 and T2-T3 were not significantly different between postures (see Table 2). Cortisol baseline levels at T1 did not significantly differ between postures ($t(74) = 0.95$, $p = 0.346$, $d = 0.22$, [−0.14–0.40]).

## Testosterone

Levels of testosterone also decreased throughout the experiment ($F(2, 146) = 19.76$, $p < 0.001$, $\eta_p^2 = 0.21$, $\eta_G^2 = 0.03$) with no different changes as a function of posture (time * posture: $F(2, 146) = 1.09$, $p = 0.340$, $\eta_p^2 = 0.01$, $\eta_G^2 = 0.00$), and no main effect of posture ($F(1, 73) = 0.13$, $p = 0.721$, $\eta_p^2 = 0.00$, $\eta_G^2 = 0.00$). The decrease over time was significant

from the first to the second ($t(74) = -3.53$, $p = 0.001$, $d_z = -0.41$, 95% CI [−0.15 to −0.04]), as well as the second to the third time point ($t(74) = -3.19$, $p = 0.002$, $d_z = -0.37$, 95% CI [−0.11 to −0.03]). Again, the decrease was similar in both posture groups (see Table 2). Testosterone baseline levels did not differ significantly between the groups ($t(73) = 0.83$, $p = 0.411$, $d = -0.19$, [−0.27–0.11]).

### Progesterone

As with the two other hormones, progesterone levels declined over time ($F(2, 142) = 33.07$, $p < 0.001$, $\eta_p^2 = 0.32$, $\eta_G^2 = 0.06$) in the same manner in both posture groups (time ∗ posture: $F(2, 142) = 0.04$, $p = 0.965$, $\eta_p^2 = 0.00$, $\eta_G^2 = 0.00$). There was no general difference between the two postures ($F(1,71) = 2.52$, $p = 0.117$, $\eta_p^2 = 0.00$, $\eta_G^2 = 0.03$). Declines between both pairs of time points were significant (T1-T2: $t(72) = -4.63$, $p < 0.001$, $d_z = -0.54$, 95% CI [−0.31 to −0.12]; T2-T3: $t(72) = -3.92$, $p < 0.001$, $d_z = -0.46$, 95% CI [−0.21 to −0.07]). As for the other hormones, this decrease was not different between posture groups (see Table 2). Progesterone baseline levels were not significantly different between the two postures ($t(71) = 1.52$, $p = 0.132$, $d = 0.36$, 95% CI [−0.06–0.48]).

### Self-report questionnaires

Participants from the two posture groups did not rate themselves as significantly different on self-esteem ($t(77) = -0.73$, $p = 0.469$, $d = -0.16$, 95% CI [−2.89–1.34]), trait anxiety ($t(77) = 0.02$, $p = 0.99$, $d = 0.00$, 95% CI [−3.82–3.88]), behavioral activation ($t(77) = -0.15$, $p = 0.88$, $d = -0.03$, 95% CI [−2.06–1.77]), and inhibition ($t(77) = 0.58$, $p = 0.562$, $d = 0.13$, 95% CI [−1.05–1.92]) prior to the testing day, nor on state anxiety at the beginning ($t(79) = 0.40$, $p = 0.689$, $d = 0.09$, 95% CI [−2.68–4.04]) or trait dominance at the end of the experiment ($t(79) = -0.90$, $p = 0.372$, $d = -0.20$, 95% CI [−3.37–1.28]).

## DISCUSSION

The present experiment investigated whether adopting expansive and constrictive postures, associated with high and low social power, respectively, impacts salivary levels of hormones related to power, stress and affiliation. Prior to our study, only one out of five studies had observed significant posture effects on testosterone and cortisol, and none had investigated effects on progesterone. Several factors had been raised as explanations for why initial findings of *Carney, Cuddy & Yap (2010)* for testosterone and cortisol did not replicate. Our design met most of the conditions which *Carney, Cuddy & Yap (2015)* suspected to be necessary for observing postural feedback effects: first, we assessed postural effects on hormones during a face categorization experiment; second, we used a cover story; third, the instructions were given by an experimenter; and fourth, participants adopted postures for maximum 2 min at a time. Moreover, following up on hypotheses raised by *Cuddy, Schultz & Fosse (2018)* and *Davis et al. (2017)*, we investigated the possibility that repeatedly holding postures over time (i.e., larger doses of posture) and/or assessing hormonal responses at longer time intervals than previous studies would induce hormonal changes.

Under these specific experimental conditions, neither testosterone and cortisol levels linked to dominance behaviors and stress reactions, nor progesterone levels related to affiliative tendencies, responded differently to adopting expansive or constrictive postures. Salivary levels of testosterone, cortisol and progesterone declined from baseline to two later post-posture samples, and did so similarly in the expansive and constrictive posture group. The first post-posture sample captured the potential incremental effect of adopting a posture twice, at approximately 23 and 11 min before sample collection. The second post-posture sample reflected the effect of adopting the same posture three times, at approximately 36, 24 and 10 min before sample collection.

Akin to four previous studies using a single posture manipulation (*Ranehill et al., 2015*; *Ronay et al., 2017*; *Smith & Apicella, 2017*; *Davis et al., 2017*), we did not replicate the effects reported by *Carney, Cuddy & Yap (2010)*. Our results add to a growing body of evidence that does not support an effect of postures on testosterone and cortisol levels. They demonstrate that even adoption of expansive and constrictive postures, for repeated but short periods of time to avoid discomfort, in between the blocks of a social task, and after providing a credible cover story, does not trigger significant hormonal changes. Thus, all the experimental characteristics listed by *Carney, Cuddy & Yap (2015)* as possible reasons for null-results in Ranehill et al.'s replication (2015) were respected in the present study. An insufficient dose of posture as well as the collection of hormone samples at inappropriate time points after the posture manipulation (see *Davis et al., 2017*) therefore seem unlikely explanations for previous non-replications. The time points at which we collected saliva samples after onset of the first posture fell into the time window (20 to 40 min) in which experimentally induced cortisol responses are strongest (*Dickerson & Kemeny, 2004*). Testosterone and progesterone responses to arousal of power and affiliation motives have been observed in a similar time window (*Schultheiss, Wirth & Stanton, 2004*; *Seidel et al., 2013*). Still, our study does not support the conclusion that power postures elicit physiological changes associated with the experience of power and stress or the need for affiliation (*Mehta & Josephs, 2011*; *Wirth, 2011*; *Schultheiss, 2013*).

Three methodological differences with previous studies merit a more detailed discussion: First, we collected three samples in total in contrast to two in all previous studies, both with a longer delay after the onset of the first posture manipulation. This procedure revealed a decline from the first to the last time point for all three hormones. This decline may either simply reflect the diurnal pattern of these hormones (*Faiman & Winter, 1971*; *Delfs et al., 1994*; *Brambilla et al., 2009*; *Liening et al., 2010*), and/or a reduction in arousal from the start to the end of the experiment as far as cortisol is concerned. Second, we examined an exclusively male sample, whereas previous studies included mostly women (with the exception of *Smith & Apicella (2017)*). If anything, this reduced variation of our dependent variables and should hence have facilitated the detection of posture effects. Moreover, in the initial study (*Carney, Cuddy & Yap, 2010*) and one of its replications (*Ranehill et al., 2015*), effects on testosterone and feelings of power were stronger in men than in women (see *Credé & Phillips, 2017*). Nevertheless, we did not observe any effect in an exclusively male sample. Third, and this is a first potential limitation of our study, hormone samples were not collected at exactly the same

time points for all participants as in previous studies, but after participants had finished a fixed number of blocks from the face categorization task at their own speed. Yet, the distribution of sampling time points was very similar in both posture groups and all samples were collected in a time window in which hormonal responses generally occur (*Dickerson & Kemeny, 2004*; *Schultheiss et al., 2012*). A second limitation is that we cannot assess whether adopting the postures was a successful power manipulation, as our study did not include any behavioral measures related to power. A final limitation concerns the size of potential posture effects on hormone levels we could detect with our sample size: all effects observed in our study were smaller than the minimal effect size of $\eta_p^2 = 0.06$ which we could have detected with 80% power. Therefore, our sample size is too small to draw any conclusions regarding potentially existing effects below this threshold.

## CONCLUSIONS

The current study assessed whether repeatedly adopting expansive and constrictive postures known as power postures induces endocrine responses that resemble the hormonal correlates of dominance and affiliative behavior. In doing so, it assessed whether larger doses of posture or collection of saliva samples at longer time intervals than previous studies would produce similar effects on testosterone and cortisol as the study by *Carney, Cuddy & Yap (2010)* in contrast to previous non-replications. Second, it investigated for the first time whether adopting postures elicits changes in progesterone levels, a hormone that seems to be associated with affiliative motives. Participants adopted an expansive or constrictive posture three times for 2 min each, in between the blocks of a face categorization task. Salivary testosterone, cortisol and progesterone levels did not differ between posture groups within a time window of 14–50 min from the beginning of the first posture. Together with results from four previous non-replications, our study thus makes it seem more unlikely that short-term manipulations of postural expansiveness or constrictiveness elicit changes of testosterone or cortisol levels, even when postures are adopted repeatedly in the context of a social task. Additionally, our results do not provide evidence for an effect of power postures on progesterone levels. Although effects on other outcome variables described as promising by *Cuddy, Schultz & Fosse (2018)* might be reproducible, the total available evidence in favor of an effect on hormones is thus currently very weak.

### Funding

This work was supported by Foundation ROGER DE SPOELBERCH, INSERM, the French National Research Agency under Grants ANR-17-EURE-0017 and ANR-10-LABX-0087 IEC and by a doctoral fellowship of the École des Neurosciences de Paris Ile-de-France and the Région Ile-de-France (DIM Cerveau et Pensée) to Hannah Metzler. The funders had no role in study design, data collection and analysis, decision to publish, or preparation of the manuscript.

## Grant Disclosures

The following grant information was disclosed by the authors:

ROGER DE SPOELBERCH, INSERM, the French National Research Agency under Grants: ANR-17-EURE-0017 and ANR-10-LABX-0087 IEC.

École des Neurosciences de Paris Ile-de-France and the Région Ile-de-France: DIM Cerveau et Pensée.

## Competing Interests

The authors declare that they have no competing interests.

## Author Contributions

- Hannah Metzler conceived and designed the experiments, performed the experiments, analyzed the data, contributed reagents/materials/analysis tools, prepared figures and/or tables, authored or reviewed drafts of the paper, approved the final draft.
- Julie Grèzes conceived and designed the experiments, contributed reagents/materials/analysis tools, authored or reviewed drafts of the paper, approved the final draft.

## Human Ethics

The following information was supplied relating to ethical approvals (i.e., approving body and any reference numbers):

The experimental protocol was approved by INSERM and licensed by the local research ethics committee (Comité de protection des personnes Ile de France III—Project C07-28, N° Eudract: 207-A01125-48) and carried out in accordance with the Declaration of Helsinki.

## Data Availability

The raw data is available at Open Science Framework, DOI 10.17605/OSF.IO/3NRSY, https://osf.io/3nrsy/

## Supplemental Information

Supplemental information for this article can be found online at http://dx.doi.org/10.7717/peerj.6726#supplemental-information.

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
