# Peer review of "Repeatedly adopting power postures does not affect hormonal correlates of dominance and affiliative behavior"

_PeerJ, doi:10.7717/peerj.6726_

## Round 0.1 · original submission · Minor Revisions

I was able to find two experts who were both impressed with your paper, but had some minor suggestions for you to address before the paper can be published. I think their suggestions will improve your paper. Please attend carefully to their suggestions and the following in your revision.

On line 68 delete "of" after "despite"

Avoid using while and since in non-temporal contexts (e.g., line 82, check throughout).

Delete the "on" after impacts on lines 137 and 310.

You discuss the effects on progesterone with both naturally occurring and experimentally induced motivations; however, you do not address the ecological validity of instructional manipulations. It seems to me there is a big difference between adopting a posture because you are instructed to do so, and adopting a posture because you are in a situation that makes you feel dominant or submissive. Thus, I'm not really convinced by the social context manipulation here. Can you provide a stronger rationale to convince me?

The measures should be clearly linked to variables identified in the Introduction. Why did you include the STAI and the BIS/BAS and RSES, for example? Make it clear what constructs are measured by the BIS/BAS. You should include reliability information for your scales, describe the scoring and provide sample items.

Move the' on line 206 to after Participants'

In the introduction, you suggest that the experimenter was naive to the posture condition but this does not seem to be the case based on your procedure.

Do you see differences between the groups if you examine T1 only?

Thank you for submitting such a nicely presented paper to PeerJ.

·

Basic reporting

The paper is well written. I only have a few minor comments. Once addressed, the paper will be ready for publication, and will make a nice contribution to the literature.

Experimental design

Rigorous investigation performed to a high technical & ethical standard.
Methods described with sufficient detail & information to replicate.

Validity of the findings

Data is robust, statistically sound, & controlled.
Conclusion are well stated, linked to original research question & limited to supporting results.

Additional comments

The paper is well written, and reports a well-designed experiment.
I only have a few minor comments. Once addressed, the paper will be ready for publication, and will make a nice contribution to the literature.

While the association between cortisol and stress is well established, I would use more tentative language when discussing the the status / affiliative effects of testosterone and progesterone, as they are based on early findings.

The reference below should be added when discussing the association between testosterone and status (as far as I know it is the only study that directly tested the hypothesis in a causal fashion).

Nave, Gideon, et al. "Single-dose testosterone administration increases men’s preference for status goods." Nature communications 9.1 (2018): 2433.

The p-curve analysis of Cuddy, Schultz & Fosse, 2018 was flawed, see http://datacolada.org/66 -- I would add a footnote about this issue and refer to the blog post.

I suspect that the bars are hiding some of the dots in Fig. 2 – this should be fixed.

Can you explain what “We used a well-established “number estimation style” procedure to induce minimal group membership, assigning participants to either the group of over- or under-estimators”: mean, and add a relevant reference?

What does “see Dotsch & Todorov, 2012 for an example of the noisy stimuli used for reverse correlation of mental representations)” mean? This seems out of place.

When reporting the results, please include 95% confidence intervals of the effects (this is especially important for a null result). Please indicate if the point estimates from Cuddy et al. (2010) lie outside the confidence intervals.

Reviewer 2 ·

Basic reporting

The article is clear and data are provided with analytic scripts.

The authors might find the special pre-registered issue on power poses in Comprehensive Results in Social Psychology (Vol 2, 2017) to be especially useful for finding additional studies that have examined the effects of power poses on a variety of outcomes for their references.

The authors should highlight the novel aspect of their work (the examination of progesterone on affiliation motives) more in their abstract and manuscript.

Experimental design

The experimental design is appropriate. However, the authors should provide less methodological details in the introduction, as this is redundant with their methods sections.

Descriptions of the timeline of the study and the power poses should be split into two figures and presented much earlier in the methods section to allow the reader to understand the study design. The authors can substantially cut their description of the poses by referencing the figure of the poses.

Results: Given that the effects of power poses are most robust on feelings of power, did the authors collect evidence of feelings of power? Did they collect any behavioral measures related to power? If so, they should be reported.

Results: I commend the authors on their open data and analytic scripts. The authors could comment their scripts more fully, though. This will help researchers who are unfamiliar with R.

Results: The authors should clarify what eta-squared G is.

Validity of the findings

My major comment has to deal with the robustness of the data (see below):

Absence of Evidence is Not Evidence of Absence:

The authors’ find no evidence that physical poses have any impact on hormone levels, regardless of the “dose” of those poses. They interpret this as evidence that poses do not impact hormone levels. For example, they state (in the discussion):

“While effects on other outcome variables described as promising by Cuddy et al. (2018) might be reproducible, the available evidence against an effect on hormone samples begins to CLEARLY outweigh evidence for such an effect.” (emphasis added)

However, the frequentist hypothesis-testing framework does not allow this conclusion. If the authors want to make this strong conclusion they need to use a statistical approach that can determine equivalences (e.g., equivalence testing; Lakens, 2017) or use a Bayesian approach.

This is particularly important because the authors likely do not have the data necessary to conclusively argue that poses do not impact hormone levels. With only 82 participants, the precision in their estimates in hormone levels is low, which will render such results inconclusive. I do not think that the authors need to show conclusively that the data are more likely under a null model (e.g., the Bayesian approach). Rather, I think they should be more circumspect in their conclusions.

This is related to the authors’ power analysis. They note that they based their sample size off the smaller effect size from Carney et al. (r = .34; 2010). According to this benchmark, their sample of 82 would have 90% power to detect this effect. However, r = .34 is a relatively large effect size for social/personality psychology. My expectation is that if physical poses impact hormonal correlates of power, that the effect would be quite small, say, r = .10. This more conservative estimate would be more in line with the effects of the most robust effect of physical poses—on feelings of power. Using this benchmark, the authors only have 15% power to detect this effect. This level of power is far too low to argue that their null results are evidence for the absence of an effect. I would like them to recalculate their power analysis based on an estimate lower level of power, and then to discuss the limitations of their study for detecting small effects.

Additional comments

The authors present a single study testing the effects of physical poses on hormone levels. This study attempts to replicate, improve, and extend the results of Carney, Cuddy, and Yap (2010). The authors find no evidence that poses impact levels of cortisol, testosterone, or progesterone. These results contribute to the growing body of evidence that does not support the conclusion that hormonal changes drive the effects of physical poses on feelings or behavior. The paper is well written and I commend the authors for providing their data and analytic scripts.

---

## Round 0.2 · Minor Revisions

Thank you for your careful revision and thorough response letter.

There are a number of very minor grammatical issues that need to be corrected before the MS can be officially accepted:

Please delete the first "of" on line 70 and line 108. "of" isn't needed after "despite". Please check throughout.
Please change "which" to "that" on line 75 and 127.
Please move the "only" to after "focused on" on line 80.
Please add the ' after individuals on line 81.
Use "and" instead of "&" outside of parentheses (e.g., line 103). Please check carefully throughout.
The "do or do not" is not needed on line 122.
Write out all authors in full only the first time the paper is cited (e.g. line 144 can be Carney et al. 2015). Check throughout (line 397 also).
On line 142, change "in" to "of."
Please insert commas after i.e. and e.g.
On line 189 please change "Since" to "Because."
On line 325, insert "with" after "As."

---

## Round 0.3 · accepted · Accept

Thank you for attending to the final minor round of corrections, and thank you for sending important replication work to us.

#